# Nurses’ Well-Being at Work in a Hospital Setting: A Scoping Review

**DOI:** 10.3390/healthcare12020173

**Published:** 2024-01-11

**Authors:** Débora Almeida, Ana Rita Figueiredo, Pedro Lucas

**Affiliations:** 1Hospital Professor Doutor Fernando Fonseca, EPE, 2720-276 Amadora, Portugal; debora.almeida@campus.esel.pt; 2Nursing Research, Innovation and Development Centre of Lisbon (CIDNUR), Escola Superior de Enfermagem de Lisboa, 1600-190 Lisboa, Portugal; prlucas@esel.pt

**Keywords:** hospitals, management, nurses, review, well-being at work

## Abstract

The aim of this review was to analyze the scientific evidence about nurses’ well-being at work in the hospital context. Well-being is present in our daily experiences, whether in a personal or professional context. Nurses are frequently put under pressure and stressed at work, which can influence their well-being. Nurses’ well-being at work in a hospital setting is crucial due to its relevance to occupational health, the quality of patient care, and the identification of stress and satisfaction factors. Methods: This systematic review followed the methodological guidelines of the Joanna Briggs Institute (JBI). The databases searched included CINAHL, MEDLINE, Scopus, Cochrane Database of Systematic Reviews, LILACS, Scientific Electronic Library Online (SciELO), and the Open Access Scientific Repositories of Portugal (RCAAP). The following inclusion criteria were defined: studies in Portuguese or English; with abstracts or full texts available; with a publication date from 2018 to 2022; and research containing the identified keywords in the title (TI) or abstract (AB). To organize and synthesize the data, we used a table to extract the significant information from each included study. Results: Eight studies were included, all of them emphasizing the assessment of well-being at work and the manager’s intervention to promote this well-being. We found that most studies indicate that nurses are satisfied with their work. There are several factors that can influence this positive level of well-being at work, such as interaction with management, culture, and organizational commitment. It can be concluded that nurse managers have a decisive role in promoting well-being at work.

## 1. Introduction

### 1.1. Well-Being at Work

Well-being is a positive state experienced by individuals and societies. It is a resource for daily life and is determined by social, economic, and environmental conditions [1]. There are several models related to well-being at work, including the Ryff model, which proposes that well-being includes six dimensions: self-acceptance, positive relationships with others, autonomy, environmental mastery, life purpose, and personal growth; Maslow’s theory of self-actualization, which suggests that well-being is achieved when basic needs are met and the individual has the opportunity to develop fully; and Deci and Ryan’s theory of self-determination, which indicates that well-being is achieved when needs for autonomy, competence, and social relatedness are met [2].

Well-being is a comprehensive concept that extends beyond physical health, encompassing job satisfaction and overall quality of life. This concept is influenced by social factors and is not confined to the workplace, being considered a crucial determinant of productivity at individual, enterprise, and societal levels. Furthermore, the importance of safety and health aspects at work stands out as an integral part of workplace well-being. Recognizing and promoting well-being at work is essential for creating healthy and efficient work environments [3].

Well-being at work is a multidimensional phenomenon influenced by several factors of professional life, such as the quality of life, safety of the physical and psychosocial environment, climate, and work organization [4]. Well-being can be interpreted as a happy state of contentment, low levels of distress, and intact physical and mental health, influencing the way people manage their daily lives [5].

Well-being at work can be translated as a set of cognitive and affective aspects. Cognitive aspects refer to general job satisfaction with other employees and with their salary. The affective aspects are related to emotional responses (feelings such as anxiety, enthusiasm, depression, and comfort) directed at the work context [6].

It is understood that the definition of well-being at work differs between organizations and countries. For nurses, well-being at work is one of the most important factors in the profession. Elements such as a balance between professional and personal life, satisfaction, commitment, and autonomy are essential and must be promoted [5,7].

When well-being at work is evaluated as positive, employees are more satisfied, and organizations have a higher success rate [8]. Therefore, it is very important to carry out this study since only by understanding the level of well-being of professionals can we find strategies to maintain or optimize it.

### 1.2. Well-Being at Work and Nursing

Nurses face high levels of stress and emotional pressure daily, which directly impacts their health and reduces their motivation to work [9]. Job satisfaction, a facet of job well-being, is fundamental not only for nurses but also for organizations, as it has a direct effect on nursing performance in any environment in which nurses work [9].

Among the characteristics related to nurses’ well-being at work are working time, satisfaction, support from managers and colleagues, appreciation, respect, balance between professional and personal life, organizational culture, an insufficient number of nurses, work and psychological pressure, a low variety of tasks, role conflicts, little autonomy, and a poor relationship between doctors and nurses [7,10]. 

It is also considered that well-being at work includes the satisfaction of the desires and needs of employees through the performance of their role in the organization. The more a person feels valued, recognized, and autonomous, with growth expectations, with environmental support, financial resources, and pleasure in belonging to the organization, the greater the level of well-being they can feel [11].

When there is a high level of well-being at work, there are positive outcomes for enthusiasm, efficiency, and satisfaction in nursing. On the other hand, a low level of well-being at work leads to professional exhaustion and increased conflict among employees [11].

Managing burnout and increasing commitment at work may be crucial to deterring nurses from considering leaving the profession [12]. It is important for organizations to improve the work environment and provide support to professionals in order to prevent burnout and promote the health and well-being of their employees [13,14]. Organizations with employees with organizational commitment have better quantitative and qualitative results and report greater satisfaction, health, and well-being among their users and employees [15]. 

There is growing awareness of the importance of well-being at work and how to act to improve it; therefore, the study of well-being at work is in a growth phase, and it is important to carry out new investigations into this area [6].

The promotion of well-being at work is extremely important, and organizations should promote a favorable work environment to minimize stressful situations [8,16].

In this way, organizations need to promote the feeling of belonging to the organization in favor of worker satisfaction, motivation, health, and increased productivity [17]. Several organizations use strategies to underline that they value their employees, increasing job satisfaction and well-being [6].

The satisfaction of health professionals has a direct influence on the quality of care provided and, at the same time, reduces burnout, turnover, and absenteeism [18].

Burnout, turnover, and absenteeism are related to the nursing practice environment. Nursing practice environments with high workloads, a lack of support and resources, and stressful situations can contribute to the development of burnout among nursing professionals. Furthermore, nursing practice environments that do not offer adequate support and demonstrate a lack of professional recognition can contribute to increased turnover among nurses. All of this can culminate in higher levels of absenteeism [18]. The nursing practice environment is characterized as a set of particularities that favor or diminish professional practice [17]. These particularities can be the adequacy of human and material resources, the effective participation of nurses in health policies, the existence of scientific evidence in the care provided, the management and leadership of nurses, and good relations between different hospital groups, for example, the relationship between doctors and nurses [19].

Investing in the hospital practice environment can be seen as a low-cost strategy for safety and quality of care, increasing customer satisfaction. However, it is important to realize that improving a care delivery system not only includes making changes within the organization but also making the most effective use of human resources [9,20]. Understanding the environment is fundamental for nursing. Nurses must consider the environment in their care interventions, ensuring that there is personalized care for each user. It is important to assess the environment to identify factors that may affect the user’s health and influence nursing practice. In addition, a holistic and integrated approach to nursing is needed that considers the physical and social environment, in addition to other dimensions of the human being [21,22].

The environment of nursing practice is an influential factor that has a great impact on nursing outcomes and on the perception of quality of care and user safety. Less favorable nursing practice environments have consequences for the use of nurses’ skills and knowledge, causing professional dissatisfaction, recruitment difficulties, burnout, and turnover, with negative consequences for organizations [21,22,23].

Nurses’ satisfaction is influenced by the nursing practice environment and the role of the nurse manager, who is responsible for generating positive results among nurses, who improve their care practice, and, consequently, the users [24].

A positive nursing practice environment optimizes the health and well-being of nurses, promoting the quality of results and organizational performance [5].

A nursing practice environment with deficiencies makes it difficult to provide nursing care and to use nurses’ skills and knowledge. In addition, it leads to professional dissatisfaction, difficulty in recruiting, burnout, and turnover, with harmful consequences for institutions [23]. 

Positive nursing practice environments ensure a significant impact on the level of quality and safety, on the well-being of professionals, the motivation and commitment of health professionals, and the productivity and effectiveness of services, organizations, and health systems [22,23]. It is important to understand the structure, culture, and organization of hospitals and how these affect the results of users and nursing, since a favorable nursing practice environment is decisive for the satisfaction, maintenance, and results of users [22].

A good nursing administration with a higher quality standard promotes positive change and greater customer satisfaction [25].

The presence of a great nursing manager tends to contribute to a positive practice. Their skills are needed to develop and support an environment favorable to clinical practice, promoting cohesion at the workplace [23]. In addition, the skills of the lead nurse are part of advanced clinical practice, with greater performance that will lead to increased nursing care, health care, and increased user safety [26]. The nurse manager is responsible for creating more opportunities, resources, and support in the institutions [24]. In this way, nurse managers have an obligation to understand not only their service but to have a broader view, enriching their perspectives, increasing their knowledge, and becoming more effective as leaders [27].

Nursing management is an area of great importance because, as a leader, the nurse manager’s role is to promote a balanced workplace that can influence their collaborators to achieve goals. It can be said that the secret of well-being at work is leadership [18]. 

The organization’s management must be alert to the weaknesses of its teams, trying to identify them [14]. 

Nurse managers are responsible for understanding the reality experienced during the provision of care and finding organizational models to maximize nursing resources, guarantee greater safety and higher quality care, and achieve results [25].

### 1.3. Rational and Objectives

The strategic plan of the International Council of Nurses 2019–2021 shows the mission to represent nursing worldwide, advance the nursing profession, promote the well-being of nurses, and advocate for health in all policies [28]. Interest in the subject of well-being at work is growing, and there are several studies on satisfaction in nursing, but few analyze well-being at work and its impact on professionals [6,29]. For that, new investigations in the area are encouraged.

Well-being at work for nurses in a hospital setting is crucial due to its relevance to occupational health, the quality of patient care, and the identification of stress and satisfaction factors. The findings can inform human resource policies, recognize the significance of the nursing role, and contribute to the scientific literature. Furthermore, this review serves as a foundation for future interventions aimed at improving nurse well-being and fostering healthier and more sustainable work environments. 

The aim of this study was to analyze the scientific evidence about nurses’ well-being at work in the hospital context, since only by understanding their well-being is it possible to find improvement strategies.

## 2. Materials and Methods

### 2.1. Design 

In conducting this systematic review on nurses’ well-being at work in hospital settings, we adopted a methodological approach that combined elements of the Joanna Briggs Institute (JBI) with the guidelines of the Preferred Reporting Items for Systematic reviews and Meta-Analyses extension for Scoping Reviews (PRISMA-ScR). The choice to integrate these specific approaches provided a good framework that allowed us to comprehensively address the complexity of the topic in question. The JBI guidelines provided a rigorous methodological framework for the search, selection, and quality assessment of included studies, while the PRISMA-ScR principles improved transparency and clarity in the presentation of results [30].

### 2.2. Eligibility Criteria

For this study, all primary and secondary studies, as well as opinion articles and reflections, were considered. To carry out the research, the search terms in the title and abstract were considered. The following inclusion criteria were defined: studies in Portuguese or English; with abstracts or full texts available; with a publication date from 2018 to 2022 (to map the most recent literature). 

For this review, we adopted an inclusive approach, incorporating all relevant forms of available literature. This included primary research studies that provide original data and systematic reviews that synthesize existing evidence in a comprehensive way. The inclusion of all these sources of evidence was grounded in the search for a holistic understanding of the topic, allowing us to address the complexity and diversity of perspectives present in the available literature. It is important to note that, in order to ensure a comprehensive and inclusive approach, no studies were excluded during the selection process. All studies identified through the search strategy were considered and assessed for relevance to the objectives of this review.

### 2.3. Sources Information and Seach Strategies

The review question was formulated based on the PCC strategy, which considered the following: population (P), nurses; concept (C), well-being at work; and context (C), hospital.

The guiding question is: “how is the well-being at work of nurses in the hospital characterized in the scientific evidence?”. Regarding the search strategies, the keywords identified in Health Sciences Descriptors (DeCS) and Medical Subject Headings (Mesh) were analyzed. It is important to understand that descriptor terms (controlled language) and terms that are not descriptors (natural language) were used. To carry out the research, the following descriptors were used: nurses; nursing; job satisfaction; hospitals. In addition, the following free terms were used: well-being at work; work environment. The Boolean operators used were AND and OR.

The keywords used initially in English were: (nurses OR nursing OR nurs *) AND (well-being at work OR job satisfaction OR work environment) AND (hospitals). Similar keywords were used for all the data in the databases. 

Three stages of research were carried out. The first was developed in CINAHL and MEDLINE, with an analysis of studies containing the identified keywords in the title (TI) or abstract (AB). This first phase of the scoping review plays a crucial role in the initial identification of relevant studies and in defining parameters for the subsequent phases of the review. After this, the same keywords and search terms were used in the remaining databases: Scopus; Cochrane Database of Systematic Reviews; LILACS; Scientific Electronic Library Online (SciELO); and the Open Access Scientific Repositories of Portugal (RCAAP).

In the third stage, we tried to find new studies, which were identified by searching the bibliographic references of all included articles, following a snowball manual search strategy [30]. Table 1 shows the search strategies used in each database.

In the articles where there was doubt, after analyzing the abstracts, they were extracted and read in full, and, after reading them, if they answered the question initially proposed, they were selected to integrate this scoping review.

Two researchers carried out the selection process. There were no disagreements, so there was no need for a third researcher.

This scoping review adhered to the Preferred Reporting Items for Systematic reviews and Meta-Analysis Extension for Scoping Reviews (PRISMA-ScR) guidelines. The PRISMA-ScR checklist guided the search strategy, establishing inclusion and exclusion criteria and influencing the assessment of the methodological quality of the included studies. The methodology adopted was based on the PRISMA-ScR recommendations to ensure consistency, transparency, and rigor in the review process.

According to the JBI scoping review methodology, data were extracted from the articles included in the review using a results extraction table, according to the review objective and question [30].

Following the search, all identified citations were collected and uploaded into Mendeley V1.19.8 (Elsevier, Amsterdam, The Netherlands), and duplicates were removed.

### 2.4. Selection Process

A total of 161 articles were identified in the selected databases. Of these, eight articles were removed for being duplicates, leaving a total of 153 articles. 

After reading the titles of the 153 articles, 126 articles were excluded because they did not contain the search terms, leaving a total of 27 articles. Next, the abstracts of the 27 articles were read; nine articles were excluded for their abstracts, leaving a total of 15 articles. 

Once the full-text articles were obtained, they were read and examined. Eight articles met the inclusion criteria, of which seven were applied to nurses in a hospital setting and one was a systematic review. No studies were added after analyzing the bibliographic references of the selected studies. 

### 2.5. Data Collection Process

To organize and synthesize the data, we employed a table to extract significant information from each included study. This approach provided a clear view of participant characteristics, study designs, and key results, facilitating the analysis and interpretation of the findings. The data extraction table was organized according to the following data: lauthor(s); year of publication; country; participants; results and conclusions.

## 3. Results

Figure 1 specifies the results of the analysis steps, following the PRISMA flowchart model.

The articles included in this scoping review covered the responses of 3100 nurses who develop their activity in the hospital context. These studies are mostly from Europe, corresponding to 50% of the studies, with 12.5% of studies from Australia (represented by Melbourne) and 25% from Asia (two studies from Saudi Arabia and China). A study was found that compares nurses from Chile and Spain, corresponding to the remaining 12.5%.

A cross-sectional investigation with a quantitative approach was used in 87.5% of the studies. In addition, a systematic review of the literature was found.

By analyzing the results of the eight articles, two thematic categories emerged: evaluation of occupational well-being and the implication of the nurse manager/leader to improve well-being [30]. Table 2 shows the characteristics of the studies analyzed.

## 4. Discussion

The aim of this scoping review was to analyze the scientific evidence about the well-being of nurses in the workplace. To respond to this objective, seven primary studies and a systematic review of the literature were included.

Contrary to traditional psychology, positive psychology emphasizes positive life factors, such as well-being at work. Work-related stress is an important aspect of nursing as it influences well-being at work and has been reported to have long-term consequences [14].

Employees with positive work-related characteristics are likely to have greater performance, satisfaction, commitment, and well-being at work [35].

The promotion of well-being at work is extremely important, and organizations should promote a favorable work environment to reduce the number and/or intensity of stressful situations their employees are involved in [7,16].

The characteristics for achieving well-being at work that employees appreciate are respect, support, and achievement of objectives [32]. Nurses demonstrate that the organizational characteristics of their workplace, the social valuation of the position, and the recognition of their efforts to be more successful bring them greater satisfaction and, consequently, a greater level of well-being at work [16].

Several organizations use strategies to underline that they value their employees, increasing job satisfaction and well-being [6]. 

The more nurses feel empowered and involved in the decision-making process, the more the practice environment is favorable for them to exercise autonomy and all their skills. In addition, it is essential that nurse managers foster and promote environments that favor high-quality care, as they will bring greater professional satisfaction and better results for clients [35].

It is very important that nurses receive recognition for their efforts for the organization and are encouraged to be more successful, bringing greater satisfaction and a higher level of well-being at work [16].

Nurses are generally satisfied with their work. In addition, there are several factors that can influence the well-being of nurses at work, such as interaction with management, culture, and organizational commitment, and nurse managers have a decisive role in promoting well-being at work [32].

Health organizations are complex systems that require balanced management between the needs of users, the institution itself, and professionals [19]. Institutions must find strategies and ways of compensating professionals based on a solid organizational culture and leadership, which in turn promotes effectiveness, efficiency, professional satisfaction, and, consequently, well-being at work [19].

### 4.1. Well-Being at Work

In general, nurses evaluate their well-being at work very positively, feeling satisfied with their work [8,14].

However, when evaluating the factors related to this well-being at work, such as the interaction with the head nurse, organizational culture, the interaction between personal and professional life, and organizational commitment, significant differences are found [14].

Nurses’ levels of well-being at work varied significantly in terms of education level, level of job satisfaction, and level of life satisfaction [8]. In one of the analyzed studies, 52% of the nurses noted that they were frequently exposed to stress in the workplace, and 30% were constantly exposed to stress in the workplace [8].

The characteristics of well-being at work, namely the number of working hours, job satisfaction, and support from managers and colleagues, were highlighted by nurses, relating them to their well-being at work. In addition, the elements related to well-being at work were outlined as follows: feeling valued, respect, support, work–life balance, and organizational culture [7].

In another study, a high level of well-being was demonstrated (about 83%), indicating that there is a positive relationship between well-being at work, communication, emotional intelligence, and the ability to empathize [34].

In the systematic review, most of the studies included found a positive response in relation to the well-being of newly graduated nurses. The authors consider that well-being at work is related to structural empowerment and career satisfaction [33]. It is noteworthy that the level of exhaustion and stress decreases over time as the professional satisfaction of nurses increases [32].

Nurses demonstrate that the organizational characteristics of their workplace, the social appreciation of the position, and the recognition of their efforts to be more successful bring greater satisfaction and, consequently, greater well-being at work [16].

In another study, 53.6% of nurses reported low levels of well-being. This low level of well-being can be explained by the relevance of the topic under study, related to COVID-19, demonstrating the growing challenges that nurses faced during the crisis, such as concern for their safety and personal stress [31]. In the same study, it is highlighted that nurses reported several ethical dilemmas, safety fears, consequences of stress, and low well-being [34].

When assessing lower levels of well-being at work, it is highlighted that this may be related to burnout, emotional exhaustion, and stress associated with the workload. In addition, work fatigue has direct effects on the well-being of employees and, consequently, on the care provided [31,32].

Nurses demonstrate that the organizational characteristics of their workplace, the social valuation of the position, and the recognition of their efforts to be more successful bring them greater satisfaction and, consequently, a greater level of well-being at work [32]. It is important to understand the emotions experienced by nurses through a good environment and workload reduction, trying to create a harmonious workforce [34].

Well-being is related to nurses’ motivation to work and provide high-quality care. The promotion of this motivation is included in the skills of the nurse manager [9,14].

Factors related to work well-being include education level, job satisfaction, life satisfaction, interaction with the head nurse, organizational culture, interaction between personal and professional life, and organizational commitment, with significant differences [7,14].

Nurses’ well-being at work is related to structural empowerment and career satisfaction. Thus, it is very important that nurses understand the recognition of their efforts for the organization to be more successful, bringing greater satisfaction and a higher level of well-being at work [16].

### 4.2. Implications of the Nurse Manager for Well-Being at Work

Leaders’ behaviors influence the well-being of their employees and, consequently, their ability to provide quality care to users. A relational leadership style, including transformational, resonant, empowering, authentic, and supportive, increases levels of employee well-being [31].

All selected studies highlight the importance of manager intervention to improve the well-being of their employees. It is up to the nurse managers to be concerned with the maintenance and durability of the same with their employees [16]. 

It is understood that stress is related to well-being at work, and the fact that stress is reduced by nursing management has a direct impact on well-being at work [31]. For a nurse manager to become effective, they must foresee the constant changes within the organization and in the environment that surrounds them and get to know their collaborators, adapting to the leadership process [7].

Nursing managers should discuss ethics and morals in nursing to prevent disorders in nurses, which can affect their willingness to work and, consequently, their well-being [14].

Interventions aimed at improving nurses’ skills are beneficial since employees with positive work-related characteristics are likely to have greater performance, satisfaction, commitment, and well-being at work [14].

The organization’s management must be alert to the weaknesses of its teams and must try to identify them. Conversations about culture and the organizational climate should be frequent; only in this way is it possible to implement strategies to increase the well-being of nurses at work [12].

Managers must consider that they will only be able to achieve positive results if their employees are satisfied with their work and feel a high level of well-being at work [8,36]. Managers must be concerned with ensuring that nurses and other employees are satisfied and healthy, intervening when this does not happen [8,32].

Several organizations use strategies to underline that they value their employees, increasing job satisfaction and well-being [6].

Nurses were critical of interactions with superiors, stating that they rarely supported them, were willing to listen to their problems, or appreciated their accomplishments at work [14]. The authors suggest that an open-door policy be applied, improving effective communication, and recognizing the nurse’s effort as an active participant in the health team [33].

Hospitals must bear in mind that they will only be able to achieve positive results if their employees are satisfied with their workplace and feel a good level of well-being. In this way, managers should be concerned with ensuring that nurses and other employees are satisfied and healthy, thus being able to intervene when this does not happen [7,8].

The management of organizations should discuss ethics and morals in nursing to prevent disorders in nurses, which can affect their willingness to work and consequently their well-being [14,34].

Hospital managers must take the initiative to encourage emotional intelligence in nurses, carrying out psychological training to improve empathy, communication, and, ultimately, their well-being at work [34].

Both management and leadership play a key role in promoting favorable working conditions, such as the implementation of care initiatives, strategies, and practices. The nurse managers are responsible for developing dynamics to contribute to nurses’ commitment to improving the quality of care and evidence-based practice [16]. 

Nurse managers have a preponderant role in the well-being of nurses since leadership affects the well-being of nurses [36]. They have a key role in promoting the well-being of their employees through the application of policies and programs that address the health needs of nurses, seeking to significantly improve the quality and safety of care provided to users. It is also their responsibility to understand what challenges their teams face and seek measures to try to solve them, namely through actions that demonstrate appreciation for their profession.

They must know and understand their employees and look for ways to invest in their growth and continuous development, either individually or as a group. It is concluded that the promotion of well-being at work is extremely important since nurses are increasingly exposed to stress and burnout. It is up to the nurse managers to be concerned with the maintenance and durability of their employees [16].

## 5. Conclusions

The scientific evidence about well-being at work is scarce. While recognizing the role of nurse managers in fostering a positive work environment, it is evident that more research is needed. The empirical studies addressing various facets of well-being at work emphasize the imperative for further comprehensive investigations. Bridging this knowledge gap will not only deepen our understanding but also inform evidence-based practices, contributing to the development of targeted strategies to enhance the overall well-being of nurses in hospital settings. Most studies indicate that nurses are satisfied with their work, answering our research question. 

There are several factors that can influence this positive level of well-being at work, such as interaction with management, culture, and organizational commitment. 

Nurse managers play a decisive role in promoting well-being at work. They should aim to understand the importance of occupational well-being during their leadership, promoting a favorable nursing practice environment wherein their employees feel valued, motivated, and supported, improving performance during their care. In addition to understanding the significance of occupational well-being throughout their leadership, nurse managers should take a proactive approach to creating a practice environment that not only meets operational demands but also nurtures and strengthens the nursing team. This involves addressing tangible issues such as workload and physical conditions, as well as cultivating an organizational culture that values mutual respect, open communication, and the formation of strong interpersonal bonds. Recognizing the positive impact of emotional support and acknowledgment on the performance of nursing professionals, managers should strive to create opportunities for professional development, celebrate individual and collective achievements, and provide ongoing support to address challenges inherent in the profession. Ultimately, effective leadership in nursing not only enhances the well-being of nurses but also translates into improved quality care for patients, solidifying the importance of a holistic approach to management in the hospital setting. 

It is important to recognize the need for further research studies to gain a deeper understanding of the well-being of nurses in the workplace as well as to identify effective strategies that can be implemented to enhance their well-being. By doing so, we can work towards improving job satisfaction among nursing professionals, which will ultimately lead to higher quality care and enhanced safety for patients.

### Limitations and Future Prospects

Due to the specificity of the topic of well-being at work, a limitation of this study was time and language, which led to difficulties in finding more studies on the subject.

## Figures and Tables

**Figure 1 healthcare-12-00173-f001:**
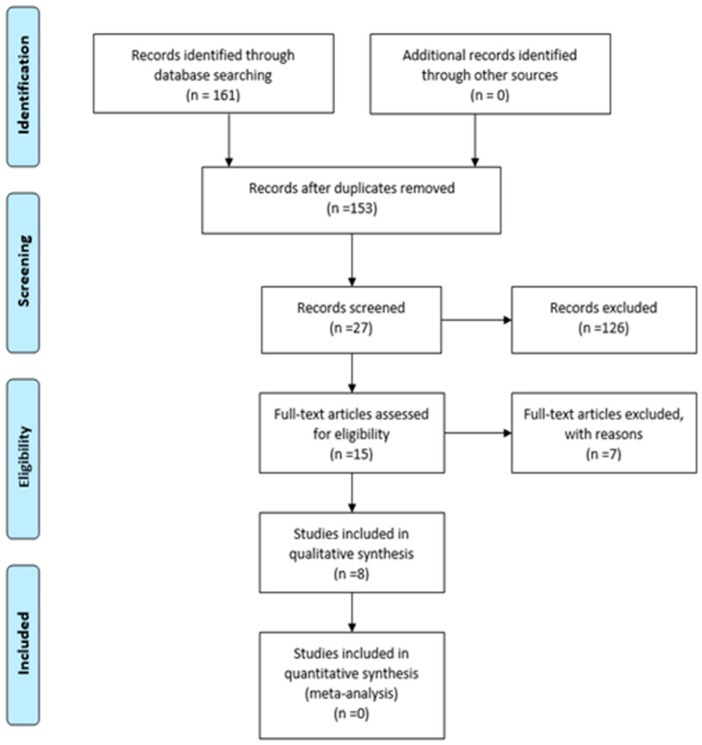
The PRISMA flowchart diagram.

**Table 1 healthcare-12-00173-t001:** Search strategies used in each database.

Database/Results	Seach Terms Used	Search Date
CINAHL (70)	(nurses OR nursing OR nurs *) AND (well-being at work OR job satisfaction OR work environment) AND hospitals	7 January 2022
MEDLINE (87)	(nurses OR nursing OR nurs *) AND (well-being at work OR job satisfaction OR work environment) AND hospitals	7 January 2022
Scopus (4)	(nurses OR nursing OR nurs *) AND (well-being at work OR job satisfaction OR work environment) AND hospitals	7 March 2022
Cochrane Database of Systematic Reviews (0)	(nurses OR nursing OR nurs *) AND (well-being at work OR job satisfaction OR work environment) AND hospitals	7 March 2022
LILACS (0)	(nurses OR nursing OR nurs *) AND (well-being at work OR job satisfaction OR work environment) AND hospitals	7 March 2022
SciELO (0)	(nurses OR nursing OR nurs *) AND (well-being at work OR job satisfaction OR work environment) AND hospitals	7 March 2022
RCAAP (0)	(nurses OR nursing OR nurs *) AND (well-being at work OR job satisfaction OR work environment) AND hospitals	7 April 2022

**Table 2 healthcare-12-00173-t002:** Characteristics of the reviewed studies.

Author(s); Year	Country	Study Design	Study Population	Sample Size	Results	Conclusions
Jarden, et al., 2018 [7]	New Zealand	Cross-sectional	Intensive care nurses	65	The most frequently cited elements included workload, job satisfaction, feeling valued, respect, support, work–life balance, and workplace culture.	Unique conceptions of well-being at work were identified. Workload and work–life balance were central characteristics. Feeling valued and experiencing respect and support were considered the most important. Critical care nurses’ conceptions of well-being at work are fundamental for future measures of well-being at work and future studies and intervention initiatives.
Konttila, et al., 2019 [14]	Finland	Cross-sectional	Nurses in psychiatric outpatient clinics	181	The nurses reported well-being at work in a very positive way but were more critical regarding the interaction with immediate superiors, organizational culture, interaction between work and private life, and organizational commitment. Experience working in psychiatric nursing and experiencing harassment have been identified as strong predictors of well-being at work.	The management of healthcare organizations should discuss nursing ethics and morals more, as well as pay attention to the ethical environment, to prevent moral distress among nurses. Several weaknesses seem to exist, mainly in the management of psychiatric outpatient clinics, that influence the well-being of nurses at work. Identifying these can help organizations develop management strategies and implement interventions to increase nurses’ well-being at work. Conversations about management culture and collegial climate should also emerge at the organizational and unit levels.
Lorber, Treven, Mumel, 2019 [3],	Slovenia	Cross-sectional	Nurses working in Slovenian hospitals	640	The nurses self-rated their satisfaction and well-being as moderate. In total, 47% of nurses were satisfied with their work, 49% rated their psychological well-being as good, 52% were frequently exposed to stress in the workplace, and 30% were always exposed to stress in the workplace. Levels of job satisfaction, psychological well-being, and subjective well-being differed significantly according to the level of education.	Nurses are moderately satisfied with their work and lives and have moderate levels of psychological and subjective well-being. Hospitals can be successful and achieve the organization’s objectives if their employees are satisfied with their work and enjoy good levels of well-being. Hospital management must recognize the importance of ensuring that nurses and other staff are happy and healthy.
Tcheco, et al., 2020 [16]	Hungary	Cross-sectional	Professionals working in nursing positions	581	Nurses present an unfavorable picture in the dimensions of depersonalization and emotional exhaustion. On the other hand, the average value of personal achievement was higher among them, which means that the interviewees feel their efforts are more successful and more positive at work. The organizational characteristics of the workplace, together with the social value of the position, significantly influence general satisfaction with life.	Promoting well-being at work is extremely important among healthcare professionals (who are increasingly exposed to stress and burnout), especially among nurses who directly participate in hospitalization.
Munoz-Rubilal, et al., 2020 [31]	Chile and Spain	Cross-sectional	Nurses	345	A total of 53.6% of nurses reported low levels of well-being; however, among those who reported low well-being, statistically significant differences were found among Spanish and Chilean nurses, as 19.4% and 37.8%, respectively, disagreed with the statement on duty of care.	Participants in both countries reported various ethical dilemmas, security fears, and consequent stress and low well-being. These results suggest that immediate actions need to be taken to address nurses’ ethics, as they may affect their willingness to work and psychological well-being.
Jarden, et al., 2021 [32]	Australia	Systematic review	---	----	The majority of studies reported greater job satisfaction over 12 months. Burnout levels were moderately high, predominantly in terms of emotional exhaustion, and stress was initially high, mainly in terms of workload, but decreased over time. Job satisfaction was positively evaluated by structural empowerment and career satisfaction.	For newly graduated nurses, levels of emotional exhaustion, workload, and stress were initially moderately high to high, decreasing over time as graduate nurses’ job satisfaction increased.
Falatah, Alhalal, 2021 [33]	Saudi Arabia	Cross-sectional	Nurses working in the healthcare system	161	Work-related stress had direct negative effects on emotional well-being. Fatigue had a direct effect on work-related affective well-being.	The Impact of stress is related to well-being at work. Reducing stress through nursing management is a fundamental element in improving work-related emotional well-being.
Li, et al., 2021 [34]	China	Cross-sectional	Registered nurses from a hospital	1475	The nurses’ well-being at work score was 83.61 ± 12.63. There was a significant positive correlation between well-being at work and communication satisfaction, emotional intelligence, and the ability to empathize.	It is recommended that hospital managers take actions to improve the emotional intelligence level of nurses and carry out professional psychological training to improve nurses’ empathy and communication satisfaction and ultimately improve their well-being at work.

## Data Availability

Data are contained within the article.

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
