# Peer review of "Nurses’ Well-Being at Work in a Hospital Setting: A Scoping Review"

_healthcare, 2024, doi:10.3390/healthcare12020173_

Round 1

Reviewer 1 Report

Comments and Suggestions for Authors

Manuscript title: Well-Being at Work of Nurses in a Hospital: A Scoping Review; requires modifications. Scoping review answers a broader question and draws a map towards future research. However, this scoping review has not provided any guidance on future research.

Abstract section is too small. Result and conclusion are not mentioned clearly.

Introduction needs to be modified. The introduction is lengthy. However, the need for review is not clearly mentioned and the aim of conducting the review is not mentioned.

In the material and method section mention how the synthesis of result was done? Were similar keywords utilized for all the databases? What about the inclusion criteria only article in English language? The material and method section has flaws and is not clear.

Results are not described and are just illustrated in the table. Authors have mentioned 153 articles and then directly 27 articles for title and abstracts this part is confusing.

Overall, this scoping review is not conducted properly and has several methodological flaws.

Comments on the Quality of English Language

Moderate editing is required. 

Author Response

Dear reviewer,

Thank you for your valuable comments/recommendations for improvement which we are very excited and honoured by. Kindly check below our responses to this issue.

Response to Reviewer 1 Comments

Point 1: Abstract section is too small. Result and conclusion are not mentioned clearly.

Response 1: thank you very much for your suggestion. Regarding results we believe that is clearer now that 8 studies were included. We added the research strategy as well as the inclusion criteria. Conclusion has been perfected.  

Point 2: Introduction needs to be modified. The introduction is length. However, the need for review is not clearly mentioned and the aim of conducting the review is not mentioned.

Response 2: thank you very much for your suggestion. We added a sub-chapter where the aim of this review is described as well as the motives to conduct this study.

Point 3: Material and methods section mention how the synthesis of the result was done? Were similar keywords utilized for all the data bases? What about the inclusion criteria only article in English language? The material and method section has flaws and is not clear.

Response 3: thank you for your question. We have perfected this area. Besides we would like to inform you that similar keywords were used for all data in the databases and added this information to the article.

Point 4:  Results are not described and are just illustrated in the table. Authors have mentioned 153 articles and then directly 27 articles for title and abstracts this part is confusing.

Response 4: thanks for your question. There were changes made to this section, we have clarified the results and added a description of how the selection of articles was made.

We take this opportunity to inform you that the paper has been revised by MDPI English editing services.

Reviewer 2 Report

Comments and Suggestions for Authors

The article addresses an important topic in health science: the well-being at work in health professionals, namely the work of nurses in the hospital environment. Individual and organizational well-being is a variable that determines the quality of the healthcare delivery, and patient safety (1). Particularly in nursing work, in the Introduction section, the authors identified literature to support the association between well-being and healthcare delivery. Facing preliminary knowledge about the main topic, the authors proposed to explore it through a literature review, which is understandable and necessary, adopting a suitable method: a scoping review. I would like to suggest changes in the Introduction (Section 1), Methodology (Sections 2 and 3), and Discussion (Section 4).

Introduction (Section 1):

There are three different topics developed in the Introduction (Section 1) that are not correctly organized. The organization should be:

Topic 1: WELL-BEING as a global concept: the authors defined this concept in lines 39-45, adopting psychological models and theories (Ryff Scales of Psychological Well-Being; The growth of self-actualization in Maslow’s theory; Self-determination theory from Edward L. Deci and Richard Ryan). The authors should initiate the introduction with these “big theories” to delimit “well-being” as a “multidimensional phenomenon influenced by several factors” (Line 1). In this regard, I suggest also adopting a global definition from WHO, described in the Health Promotion Glossary of Terms 2021 (2).

Topic 2: WELL-BEING AT WORK the authors detailed in the lines 23-38 and 56-79. In this reflection, authors adopt an approach from “managing”, “human capital”, and “human resources”, including concepts like “employee”, “organization”, “performance”, “efficiency”, “burnout”, etc. Indeed, the theories of well-being, not only from psychology but also from sociology, were adopted by human resources management (theories of human capital), to develop better conditions of work and productivity. I suggest reading the article “Well-being at Work – overview and Perspective” (3) to improve this reflection.

Topic 3 (the subject of study): WELL-BEING AT WORK OF NURSES IN A HOSPITAL that authors developed in lines 80-143. The authors initiated the reflection about the subject of study, writing: “The satisfaction of health professionals has a direct influence on the quality of care provided, and at the same time reduces burnout, turnover and absenteeism” (lines 80-81). However, the reflection is developed based on two different concepts: “nursing practice environment” (lines 82-118) and “nursing management” (lines 119-140). During this reflection, there are no more references to the concepts of “burnout, turnover and absenteeism”. Additionally, the reference to the “Strategic Plan of International Council of Nurses 2019-2021” should be adopted in the article to support the article’s objectives.

Finally, I suggest finalising the introduction with a focus on the reflection or the subject of study: what the authors want to focus on WELL-BEING AT WORK OF NURSES IN A HOSPITAL? What is the subject of study? Is the impact of the nursing practice environment and nursing management to reduce/increase burnout, turnover and absenteeism? The delimitation of the subject of study will determine the methodology description. I suggest dividing the Introduction (Section 1) into three sections: 1.1) Well-being and Work; 1.2) Well-being and Nursing; 1.3) Rational and Objectives based on the PCC question and the sub-section adopted in the discussion (section 4).

Materials and Methods (Section 2):

This section should be organized following the PRISMA checklist (http://www.prisma-statement.org/), namely: Eligibility criteria; Information sources; Search strategy; Selection process; Data collection process; Synthesis methods; Reporting bias assessment; Certainty assessment.

I have doubts about the inclusion criteria and the results of the scoping review. There are only 8 articles selected with these criteria: “studies in Portuguese or English; with abstracts or full texts available; with publication date from 2018 to 2022; that contain the identified keywords in the title (TI) or abstract (AB)” (lines 167-169)?

Results (Section 3):

Considering the Eligibility criteria, Information sources, Search strategy, and Selection process (Section 2), the Results (section 3) should include more details related to the studies that might appear to meet the inclusion criteria, but which were excluded, and explain why they were excluded.

Author Response

Dear reviewer,

Thank you for your valuable comments/recommendations for improvement which we are very excited and honoured by. Kindly check below our responses to this issue.

Response to Reviewer 2 Comments

Comments: The article addresses an important topic in health science: the well-being at work in health professionals, namely the work of nurses in the hospital environment. Individual and organizational well-being is a variable that determines the quality of the healthcare delivery, and patient safety (1). Particularly in nursing work, in the Introduction section, the authors identified literature to support the association between well-being and healthcare delivery. Facing preliminary knowledge about the main topic, the authors proposed to explore it through a literature review, which is understandable and necessary, adopting a suitable method: a scoping review. I would like to suggest changes in the Introduction (Section 1), Methodology (Sections 2 and 3), and Discussion (Section 4).

Introduction (Section 1):

There are three different topics developed in the Introduction (Section 1) that are not correctly organized. The organization should be:

Topic 1: WELL-BEING as a global concept: the authors defined this concept in lines 39-45, adopting psychological models and theories (Ryff Scales of Psychological Well-Being; The growth of self-actualization in Maslow’s theory; Self-determination theory from Edward L. Deci and Richard Ryan). The authors should initiate the introduction with these “big theories” to delimit “well-being” as a “multidimensional phenomenon influenced by several factors” (Line 1). In this regard, I suggest also adopting a global definition from WHO, described in the Health Promotion Glossary of Terms 2021 (2).

Response 1: Thank you very much for the suggestion. The article was changed, the major theories were pasted at the beginning of the introduction and, in addition, the WHO definition was introduced

Topic 2: WELL-BEING AT WORK the authors detailed in the lines 23-38 and 56-79. In this reflection, authors adopt an approach from “managing”, “human capital”, and “human resources”, including concepts like “employee”, “organization”, “performance”, “efficiency”, “burnout”, etc. Indeed, the theories of well-being, not only from psychology but also from sociology, were adopted by human resources management (theories of human capital), to develop better conditions of work and productivity. I suggest reading the article “Well-being at Work – overview and Perspective” (3) to improve this reflection.

Response 2: Thank you very much for the suggestion. We read the article and introduce some information.

Topic 3 (the subject of study): WELL-BEING AT WORK OF NURSES IN A HOSPITAL that authors developed in lines 80-143. The authors initiated the reflection about the subject of study, writing: “The satisfaction of health professionals has a direct influence on the quality of care provided, and at the same time reduces burnout, turnover and absenteeism” (lines 80-81). However, the reflection is developed based on two different concepts: “nursing practice environment” (lines 82-118) and “nursing management” (lines 119-140). During this reflection, there are no more references to the concepts of “burnout, turnover and absenteeism”. Additionally, the reference to the “Strategic Plan of International Council of Nurses 2019-2021” should be adopted in the article to support the article’s objectives.

Response 3: Thank you very much for the suggestion. Burnout, turnover and absenteeism are related to nursing practice environment. It was added to the text, lines 96-101 the relationship between the nursing practice environment and the concepts. The objective of the study was added to the introduction and related to the strategic plan reference.

Finally, I suggest finalising the introduction with a focus on the reflection or the subject of study: what the authors want to focus on WELL-BEING AT WORK OF NURSES IN A HOSPITAL? What is the subject of study? Is the impact of the nursing practice environment and nursing management to reduce/increase burnout, turnover and absenteeism? The delimitation of the subject of study will determine the methodology description. I suggest dividing the Introduction (Section 1) into three sections: 1.1) Well-being and Work; 1.2) Well-being and Nursing; 1.3) Rational and Objectives based on the PCC question and the sub-section adopted in the discussion (section 4).

Response 4: Thank you very much for the suggestion. We divided like you suggest. And introduce the aim (subject of study).

Materials and Methods (Section 2):

This section should be organized following the PRISMA checklist (http://www.prismastatement.org/), namely: Eligibility criteria; Information sources; Search strategy; Selection process; Data collection process; Synthesis methods; Reporting bias assessment; Certainty assessment.

Response 5: Thank you very much for the suggestion. We try to divided like you suggest.

I have doubts about the inclusion criteria and the results of the scoping review. There are only 8

articles selected with these criteria: “studies in Portuguese or English; with abstracts or full texts

available; with publication date from 2018 to 2022; that contain the identified keywords in the title

(TI) or abstract (AB)” (lines 167-169)?

Response 6: Thank you very much for the question. We would like to inform you that there are 8 articles that we found when applying the inclusion criteria.

Results (Section 3):

Considering the Eligibility criteria, Information sources, Search strategy, and Selection process (Section 2), the Results (section 3) should include more details related to the studies that might appear to meet the inclusion criteria, but which were excluded, and explain why they were excluded.

Response 7: Thank you very much for the suggestion. We add the information in the results

We take this opportunity to inform you that the paper has been revised by MDPI English editing services.

Reviewer 3 Report

Comments and Suggestions for Authors

Ana Rita Figueiredo et al. submitted to Healthcare a review on the well-being of nurses while working in hospital.

This manuscript has many limitations, it requires improvements and does not lend itself to a fluid reading, particularly in discussions (many sentences "start a new line" and do not provide continuity): please provide for a total semantic restructuring, as usual for scientific works.

Furthermore:

It is better to delve deeper if for the studies performed in 2020 and 2021, which fall within the enrollment criteria and which were selected, the COVID-19 pandemic may have affected the findings that emerged and in which way, explaining the limits of these studies deriving from the different investigation context.

Moreover, the Authors need to develop in the discussions the ways in which decision makers and managers (not just nurse managers) can contribute to improving organizational set-up and models and, as a cascade, the well-being of nurses and all healthcare workers.

36 citations appear in the manuscript, while 37 are cited in the references section.

Comments on the Quality of English Language

Moderate editing of English language required

Author Response

Dear reviewer,

Thank you for your valuable comments/recommendations for improvement which we are very excited and honoured by. Kindly check below our responses to this issue.

Response to Reviewer 3 Comments

Comments: The manuscript has many limitations, it requires improvements and does not lend itself to a fluid reading, particularly in discussions (many sentences “start a new line” and do not provide continuity): please provide for a total semantic restructuring, as usual for scientific works.

Response: Thank you for this suggestion. Considering this, we have improved the fluidity of the article, especially the discussion section.

Furthermore:

It is better to delve deeper if for the studies performed in 2020 e 2021, which fall within the enrollment criteria and which are selected, the COVD-19 pandemic may have affected the findings that emerged and in which way, explaining the limits of these studies deriving from the different investigation context.

Response: We thank you for this question. Regarding the studies from 2020 -2021, the aim of this study was not to focus on the covid pandemic time; but we have added a reference to this.

Moreover, the authors need to develop the discussions the ways in which decision makers and managers (not just nurse managers) can contribute to improving organizational set-up and models and, as a cascade, the well-being of nurses and all healthcare workers.

Response: Thank you for the question. As for the other hospital managers, they are mentioned throughout the results section, but we have restructured the text to make their inclusion more visible.

Comments: 36 citations appear in the manuscript, while 37 are cited in the references section.

Response: Thank you very much for the question. We have rechecked the references and corrected the flaws.

We take this opportunity to inform you that the paper has been revised by MDPI English editing services.

Reviewer 4 Report

Comments and Suggestions for Authors

Dear authors, thank you very much for the opportunity to read your work. The submitted manuscript addresses a topic of interest that is little studied. Although there are some recent systematic reviews, they do not specifically address nurses in the hospital setting. For these reasons, the scoping review design is pertinent.

To review the manuscript, I followed the PRISMA guidelines extension for scoping review.

Title: pertient and correct.

Keywords: The correct MesH descriptor for the term "Hospital" is the plural "Hospitals". "Management" is a very unspecific term as a descriptor (entering the term in the MesH Database retrieves 55 options other than "Management"). "Well-being at work" is not a MesH, the correct descriptor is "Psychological Well-being".

Abstract: The abstract should provide more information about the method, including at least: eligibility criteria, sources of evidence, charting methods.

Introduction: Correct.

Page 1 (lines 39-41) indicates that the Ryff model includes six dimensions, but only 5 are described (there must be a typo). 

Page 2 (line 82), when it describes: "According to Lake (2002)", it gives the impression that APA citation style is used (although it is understood that this is not the case, it may be confusing to indicate the date). For this purpose, it is referenced with the number [17] at the end of the sentence. Do not duplicate quote [17] in the same paragraph, just once at the end.

Page 2 (lines 88-91) the two sentences that refer to the hospital remain unconnected. It would be more appropriate to write them in a single paragraph and place it before referring to nurses (keep in mind that they talk about nurses before and after these sentences).

You must review and unify the criteria in the style to reference the text (Page 2 (line 97): [19-20] vs. page 3 (line 102): [19,20,21]).

The research question (written in literary format) or, at least, the aim should be moved to the end of the introduction. In the methods section it is correct to identify the research question in a structured way (PCC).

Methods:

Protocol and registration: Indicate whether a review protocol exists; state if and where it can be accessed (e.g., a Web address); and if available, provide registration information, including the registration number.

The method must be described in a logical and orderly manner (in my opinion it is better structured although not mandatory) following the structure (PRISMA-ScR guidelines): Design, Eligibility criteria, Information sources, Search, Selection of sources of evidence, Data charting process, Data items,  Critical appraisal of individual sources of evidence, and Synthesis of results.

In the sentence: "Three stages of research were carried out. The first was developed in CINAHL and MEDLINE, with an analysis of the keywords contained in the titles and abstracts. In the second stage, the same keywords and search terms were used in the remaining databases of the EBSCOHost platform" is not correct two stages (information sources is one stage). Do not duplicate information: CINAHL ad MEDLINE (page 4, line 154 and line 159).

The sentence: "In the third stage, new studies were identified by searching the bibliographic references of all included articles", it should be included in the search section indicating that it is a manual "snowball" search strategy.

You must distinguish that in the search strategies descriptor terms (controlled language) and terms that are not descriptors (natural language) have been used.

It is recommended to include a table or supplementary material indicating the search strategies used in each database and the search dates in each one.

The sentence: "and duplicates were removed" at the end of the methodology does not follow the logical order of the review process in the flowchart.

Results:

In the flowchart adjust the figure to PRISMA, indicating all the records that are eliminated and left in each step. How many duplicate records?, Reasons for elimination in screening by title and abstract?, Reasons for elimination in full-text screening?

Page 4 (line 103): The sentence "Once the full text articles were obtained, they were read and examined" is not relevant in results (move it to methodology where appropriate).

The sentece: "By analyzing the results of the 8 articles, two thematic categories emerged: evaluation of occupational well-being and the implication of the nurse manager/leader to improve this well-being [29]". It is not correct that you introduce this reference in the results of the study to justify how to structure the results (in two categories) without having explained this aspect in the methodology (Synthesis of results). Furthermore, this reference does not correspond to a result of the review.

Include the reference number in each of the studies in the tables. Arrange the studies in the table following a logical order (most recent publication date on top or vice versa is appropriate).

Discussion:

The discussion has been structured following criteria that do not correspond to the presentation of results.

Conclusions: It must respond to the objective. They are very extensive. It is possible that part of the conclusions correspond to discussion.

References: In general, adecuate.

Author Response

Dear reviewer,

Thank you for your valuable comments/recommendations for improvement which we are very excited and honoured by. Kindly check below our responses to this issue.

Response to Reviewer 4 Comments

Comments: Dear authors, thank you very much for the opportunity to read your work. The submitted manuscript addresses a topic of interest that is little studied. Although there are some recent systematic reviews, they do not specifically address nurses in the hospital setting. For these reasons, the scoping review design is pertinent.

To review the manuscript, I followed the PRISMA guidelines extension for scoping review. Title: pertient and correct.

Keywords: The correct MesH descriptor for the term "Hospital" is the plural "Hospitals". "Management" is a very unspecific term as a descriptor (entering the term in the MesH Database retrieves 55 options other than "Management"). "Well-being at work" is not a MesH, the correct descriptor is "Psychological Well-being".

Response 1: thank you very much for your suggestion. We correct the “hospitals”. However “Psychological Well-being" its not the definition that we want. In the search strategies we use descriptor terms (controlled language) and terms that are not descriptors (natural language).

Abstract: The abstract should provide more information about the method, including at least: eligibility criteria, sources of evidence, charting methods.

Response 2: thank you very much for your suggestion. We add the information of eligibility criteria in the abstract, sources of evidence and charting methods.

Introduction: Correct.

Page 1 (lines 39-41) indicates that the Ryff model includes six dimensions, but only 5 are described (there must be a typo).

Response 3: thank you very much. you are absolutely right, we added “Environmental Mastery”, that was missing.

Comments: Page 2 (line 82), when it describes: "According to Lake (2002)", it gives the impression that APA citation style is used (although it is understood that this is not the case, it may be confusing to indicate the date). For this purpose, it is referenced with the number [17] at the end of the sentence. Do not duplicate quote [17] in the same paragraph, just once at the end.

Response 4: thank you very much for your suggestion. We corrected the text.

Comments: Page 2 (lines 88-91) the two sentences that refer to the hospital remain unconnected. It would be more appropriate to write them in a single paragraph and place it before referring to nurses (keep in mind that they talk about nurses before and after these sentences).

Response 5: thank you very much for your suggestion. We corrected the text.

Comments: You must review and unify the criteria in the style to reference the text (Page 2 (line 97): [19- 20] vs. page 3 (line 102): [19,20,21]).

Response 6: thank you very much for your suggestion. We corrected the text

Comments: The research question (written in literary format) or, at least, the aim should be moved to the end of the introduction. In the methods section it is correct to identify the research question in a structured way (PCC).

Response 7: thank you very much for your suggestion. We corrected the text

Methods:

Protocol and registration: Indicate whether a review protocol exists; state if and where it can be accessed (e.g., a Web address); and if available, provide registration information, including the registration number.

Response 8: thank you very much for your suggestion. Unfortunately, no registration was made.

Comments: The method must be described in a logical and orderly manner (in my opinion it is better structured although not mandatory) following the structure (PRISMA-ScR guidelines): Design, Eligibility criteria, Information sources, Search, Selection of sources of evidence, Data charting process, Data items, Critical appraisal of individual sources of evidence, and Synthesis of results.

Response 9: thank you very much for your suggestion. We try to correct using PRISMA guidelines.

Comments: In the sentence: "Three stages of research were carried out. The first was developed in CINAHL and MEDLINE, with an analysis of the keywords contained in the titles and abstracts. In the second stage, the same keywords and search terms were used in the remaining databases of the EBSCOHost platform" is not correct two stages (information sources is one stage). Do not duplicate information: CINAHL ad MEDLINE (page 4, line 154 and line 159).

Response 10: thank you very much. you are absolutely right. We made corrections to the text.

Comments: The sentence: "In the third stage, new studies were identified by searching the bibliographic references of all included articles", it should be included in the search section indicating that it is a manual "snowball" search strategy.

Response 11: thank you very much for your suggestion. We add this information.

Comments: You must distinguish that in the search strategies descriptor terms (controlled language) and terms that are not descriptors (natural language) have been used.

Response 12: thank you very much for your suggestion. We add this information.

Comments: It is recommended to include a table or supplementary material indicating the search strategies used in each database and the search dates in each one.

Response 13: thank you very much for your suggestion. However, the keywords that were used are similar for all databases, information that we added in the article. Line 194 – 195.

Comments: The sentence: "and duplicates were removed" at the end of the methodology does not follow the logical order of the review process in the flowchart.

Response 14: thank you very much for your suggestion. We correct this information.

Results:

In the flowchart adjust the figure to PRISMA, indicating all the records that are eliminated and left in each step. How many duplicate records?, Reasons for elimination in screening by title and abstract?, Reasons for elimination in full-text screening?

Page 4 (line 103): The sentence "Once the full text articles were obtained, they were read and examined" is not relevant in results (move it to methodology where appropriate).

Response 15: thank you very much for your suggestion. We change this information to methodology.

Comments: The sentece: "By analyzing the results of the 8 articles, two thematic categories emerged: evaluation of occupational well-being and the implication of the nurse manager/leader to improve this well-being [29]". It is not correct that you introduce this reference in the results of the study to justify how to structure the results (in two categories) without having explained this aspect in the methodology (Synthesis of results). Furthermore, this reference does not correspond to a result of the review.

Response 16: thank you very much for your suggestion. The two categories were defined by us, taking into account the results found. That's why the reference was removed.

Comments: Include the reference number in each of the studies in the tables. Arrange the studies in the table following a logical order (most recent publication date on top or vice versa is appropriate).

Response 17: thank you very much for your suggestion. We organize the table in chronological order (from oldest to most recent) and add the corresponding reference.

Discussion:

The discussion has been structured following criteria that do not correspond to the presentation of results.

Response 18: thank you very much for your suggestion. However, this choice was made by the authors, in order to facilitate reading and presentation of the results.

Conclusions: It must respond to the objective. They are very extensive. It is possible that part of the conclusions correspond to discussion.

Response 19: thank you very much for your suggestion. We reduce the information in the conclusion and highlight the answer to our research question.

References: In general, adecuate.

We take this opportunity to inform you that the paper has been revised by MDPI English editing services.

Round 2

Reviewer 1 Report

Comments and Suggestions for Authors

Dera Authors,

All the major issues have been addressed. 

Comments on the Quality of English Language

Minor editing 

Author Response

Response to Reviewer 1 Comments

Dear reviewer,

Thank you for your valuable comments/recommendations for improvement which we are very excited and honoured by.

Just a reminder that we did MDPI's English editing services on the manuscript.

Best regards. 

Reviewer 2 Report

Comments and Suggestions for Authors

Dear authors,

Congratulations. I would like to suggest two minor changes:

- you should include Table 1 in the results.

- the limitations (in discussion) should be in the conclusion.

Author Response

Response to Reviewer 2 Comments

Dear reviewer,

Thank you for your valuable comments/recommendations for improvement which we are very excited and honoured by. Kindly check below our responses to this issue.

Topic 1: You should include Table 1 in the results

Response 1: Thank you very much for the suggestion. We have included Table 1 in the results.

Topic 2: The limitations (in discussion) should be in the conclusion

Response 2: Thank you very much for the suggestion. We added the limitations in the conclusion.

We take this opportunity to inform you that the paper has been revised by MDPI English editing services.

Best regards. 

Reviewer 3 Report

Comments and Suggestions for Authors

The manuscript has definitely improved, thanks to a partial rewriting, useful for outlining an adequate logical framework for this paper.

Author Response

Response to Reviewer 3 Comments

Dear reviewer,

Thank you very much for your comments to improve the manuscript.

Best regards. 

Reviewer 4 Report

Comments and Suggestions for Authors

Dear authors, in this new version of the manuscript, methodological issues have been improved, but some improvements can still be implemented:

The use or not of descriptors as keywords is an editorial decision for the correct indexing of the article.

In the abstract you should describe the objective after the introduction (not in the middle).

Introduction: In the new sub-heading (1.3 Rational and objectives), the objective should be described at the end of the item.

Methods:

In the section method you must start with the design heading (Scoping review according PRISMA-ScR, JBI...).

In the sentence: "...all primary and secondary studies and systematic reviews from the literature..." you should correct the duplication of information since systematic reviews correspond to secondary types of research. It would be appropriate to specify research designs included and excluded.

The following sentence must be move to the subheading search strategies: "...studies containing the identified keywords in the title (TI) or abstract (AB)".

Unify items 2.2 and 2.3 such as: Sources of information and search strategies. (the information in point 2.2 is more relevant together in the following point).

On pages 4-5 (lines 198-205) it is not relevant to separate two phases (the searches have been carried out in each of the databases mentioned in the methodology). It is correct to separate a "second" phase of searches through the references of the included studies (snowball strategy).

In the search strategies section you must specify which descriptors and free terms you used.

On page 5 (line 200) the statement: "...first phase of the systematic review..." is not correct since it is a scoping review.

Clarify whether the screening process has been performed by one or two persons (also specify how discrepancies have been resolved (third reviewer)).

You should clarify whether any critical appraisal process has been used.

Results:

Figure 1 and Table 1 are results and should be move to the results section. The sentence in page 5 (line 255) confirm this ("Figure 1 specifies the results of the analysis steps"). Figure 1 specifies the results of the analysis steps Information on the search strategies executed in each of the databases (including dates on which the searches were performed) should be available in a table or supplementary file included in the methodology section. it is a quality criterion to report the search strategies used (even if the same terms have been used in each of them).

The information in Figure 1 is not correct. The number of record outputs (duplicates, screening by title and abstract, and full-text screening) should always be noted at each step. The number of records after duplicates (n=153) and the number of records screened by title and abstract (n=27) do not match. What happened to the remaining records? 153-27=126?

Resuts section: In the first paragraph of results, the methodology should not be repeated.

Author Response

Response to Reviewer 4 Comments

Dear reviewer,

Dear authors, in this new version of the manuscript, methodological issues have been improved, but some improvements can still be implemented:

Response: Thank you for your valuable comments/recommendations for improvement which we are very excited and honoured by. Kindly check below our responses to this issue.

Topic 1: The use or not of descriptors as keywords is an editorial decision for the correct indexing of the article.

Response 1: Thank you very much for the suggestion. We had a lengthy discussion during the review and ultimately decided to include both indexed and non-indexed terms in the article. While indexed terms assist in indexering the article, non-indexed terms are equally important for perfoming a thorough search on the topic. We are confident that this combination will make the article easily searchable, while also incorporating the necessary keywords that accurately represent the article’s content.

Topic 2: In the abstract you should describe the objective after the introduction (not in the middle).

Response 2: Thank you very much for the suggestion. In the abstract we put the objective before the introduction

Topic 3: Introduction: In the new sub-heading (1.3 Rational and objectives), the objective should be described at the end of the item.

Response 3: Thank you very much for the suggestion. We put the objective at the end of the item and corrected the opening sentence

Methods:

Topic 4: In the section method you must start with the design heading (Scoping review according PRISMA-ScR, JBI...).

Response 4: Thank you very much for the suggestion. We added this information.

Topic 5: In the sentence: "...all primary and secondary studies and systematic reviews from the literature..." you should correct the duplication of information since systematic reviews correspond to secondary types of research. It would be appropriate to specify research designs included and excluded.

Response 5: Thank you very much for the suggestion. We correct the duplicate information. We specify in the text the research concepts included and excluded.

Topic 6: The following sentence must be move to the subheading search strategies: "...studies containing the identified keywords in the title (TI) or abstract (AB)".

Response 6: Thank you very much for the suggestion. We move the sentence to search strategies.

Topic 7: Unify items 2.2 and 2.3 such as: Sources of information and search strategies. (the information in point 2.2 is more relevant together in the following point).

Response 7: Thank you very much for the suggestion. We put the items together as suggested.

Topic 8: On pages 4-5 (lines 198-205) it is not relevant to separate two phases (the searches have been carried out in each of the databases mentioned in the methodology). It is correct to separate a "second" phase of searches through the references of the included studies (snowball strategy).

Response 8: Thank you very much for the suggestion. We have corrected the text.

Topic 9: In the search strategies section you must specify which descriptors and free terms you used.

Response 9: Thank you very much for the suggestion. We add this information in the text.

Topic 10: On page 5 (line 200) the statement: "...first phase of the systematic review..." is not correct since it is a scoping review.

Response 10: Thank you very much for the suggestion. It was a mistake on our part. It has been corrected in the text.

Topic 11: Clarify whether the screening process has been performed by one or two persons (also specify how discrepancies have been resolved (third reviewer)).

Response 11: Thank you very much for the suggestion. We add this information in the text.

Topic 12: You should clarify whether any critical appraisal process has been used.

Response 12: Thank you very much for the suggestion. We add the information.

Results:

Topic 13: Figure 1 and Table 1 are results and should be move to the results section.

Response 13: Thank you very much for the suggestion. We change figure 1 and table 1 to the results section.

Topic 14: The sentence in page 5 (line 255) confirm this ("Figure 1 specifies the results of the analysis steps"). Figure 1 specifies the results of the analysis steps Information on the search strategies executed in each of the databases (including dates on which the searches were performed) should be available in a table or supplementary file included in the methodology section. it is a quality criterion to report the search strategies used (even if the same terms have been used in each of them).

Response 14: Thank you very much for the suggestion. We added a table with Search strategies used in each database

Topic 15: The information in Figure 1 is not correct. The number of record outputs (duplicates, screening by title and abstract, and full-text screening) should always be noted at each step. The number of records after duplicates (n=153) and the number of records screened by title and abstract (n=27) do not match. What happened to the remaining records? 153-27=126?

Response 15: Thank you very much for the suggestion. We correct the figure 1. Also, we have added a more detailed description of the research to the chapter on the selection process. However, it should be noted that the 126 articles were excluded only by reading the title.

Topic 16: Results section: In the first paragraph of results, the methodology should not be repeated.

Response 16: Thank you very much for the suggestion. We have corrected the text.

We take this opportunity to inform you that the paper has been revised by MDPI English editing services.

Best regards